# Traditional Gender Role Attitudes and Job-Hunting in Relation to Well-Being: A Cross-Sectional Study of Japanese Women in Emerging Adulthood

**DOI:** 10.3390/ijerph22091385

**Published:** 2025-09-04

**Authors:** Yumiko Kobayashi, Yuki Imamatsu, Azusa Arimoto, Kenkichi Takase, Ayumi Fusejima, Kanami Tsuno, Takashi Sugiyama, Masana Sannnomiya, Tomoyuki Miyazaki

**Affiliations:** 1Research Department, Institute for Health Economics and Policy, Association for Health Economics Research and Social Insurance and Welfare, Tokyo 105-0001, Japan; yumiko.kobayashi@ihep.jp; 2Department of Community Health Nursing, Graduate School of Medicine, Yokohama City University, Yokohama 236-0004, Japan; 3Department of Public Health Nursing, School of Nursing and Rehabilitation Sciences, Showa University, Tokyo 142-8555, Japan; 4Center for Promotion of Research and Industry-Academic Collaboration, Yokohama City University, Yokohama 236-0027, Japan; 5Department of Psychology, Chuo University, Tokyo 192-0393, Japan; 6Department of Anesthesiology, Graduate School of Medicine, Yokohama City University, Yokohama 236-0004, Japan; 7Department of Psychology and Information Design, Kanazawa Institute of Technology, Nonoichi 921-8501, Japan; 8School of Health Innovation, Kanagawa University of Human Services, Yokohama 210-0821, Japan; 9Department of Human Sciences, Kanagawa University, Yokohama 221-8686, Japan

**Keywords:** job-hunting, well-being, gender role, emerging adulthood

## Abstract

Employment and job-hunting can improve well-being by increasing confidence among emerging adults when equal employment opportunities exist for women and men. However, the relationship between well-being, traditional gender role attitudes, and job-hunting among women in emerging adulthood remains unclear. This study examined the interactions between gender role attitudes and job-hunting in relation to the well-being of emerging adult women. An online survey was conducted in five universities and five companies in Japan. The dependent variable was well-being. The explanatory variables were job-hunting experience within the past 6 months and traditional gender role attitudes measured by the gender role stressor scale. Of the 137 women, we analyzed the data from 132 participants with no missing data. Thirty-five (26.5%) participants were employed and had job-hunting experience. Multiple regression analysis showed that job-hunting experiences were negatively associated with well-being. Additionally, gender role attitudes were not associated with well-being. In the interaction between job-hunting experience and gender role attitudes, the more traditional one’s attitude toward gender roles is, the more negative the relationship between job-hunting experience and well-being. Job-hunting may not necessarily lead to well-being for all women, so women’s attitudes toward gender roles should be considered and respected.

## 1. Introduction

Emerging adulthood refers to the period between ages 18 and 29 years, characterized by identity exploration, instability, self-focus, feeling in-between, and optimism about possibilities [1]. Arnett proposed that emerging adulthood, defined as an extended period of education, delayed marriage, or parenting, creates a new space between adolescence and young adulthood [1]. This generation considers getting a job to encompass more than just earning money; it also involves considerations of social status, quality of life, and finding meaning in life [1,2,3,4]. Previous studies have reported that employment and job search activities may improve well-being [5,6,7,8,9]. A meta-analysis of longitudinal studies and cross-country research has suggested pathways linking employment status to well-being [4,5]. These pathways primarily involve financial rewards, in addition to direct psychological influences such as social contacts and social status [5,6]. Prior research examining the potential influence of employment status on the health of young individuals suggests that stable employment is vital [10,11,12].

For young people who have completed their studies, job-hunting is an important first step toward economic independence, but employment for young people in Japan is unstable. Japan ranks in the top third of countries in the Global Youth Well-being Index regarding economic opportunities, due to its low youth unemployment rates and the few young people without education or job training [13]. However, despite the importance of employment, Japanese teenagers and people in their 20 s are likely to quit their jobs earlier than older generations, and this is especially true for women [14]. In the 2021 survey, the turnover rate for males and females aged 20–24 was 24.2% and 26.9%, respectively [14]. This trend is not exclusive to Japan; it is also prevalent in other countries [15].

### 1.1. Literature Review and Hypotheses

#### 1.1.1. Well-Being

Well-being is a particularly important aspect of development during emerging adulthood. Well-being is defined as a state composed of five dimensions: Positive Emotion, Engagement, Relationships, Meaning, and Accomplishment [16]. Positive Emotion refers to feelings such as joy or being moved, while Engagement indicates active involvement in activities [16]. Relationships concern the quality of one’s connections with others, and Meaning refers to an awareness of life’s purpose [16]. Accomplishment reflects the sense of achievement [16]. Emerging adulthood is a life stage in which individuals are more likely to encounter opportunities for positive emotions associated with life events, active involvement in academic or occupational pursuits, and the formation of new interpersonal relationships [1]. Moreover, entering the workforce is not merely about having a job; it also provides opportunities to discover meaning in life and to experience a sense of accomplishment. Therefore, job-hunting, as an activity necessary for establishing new connections with society, may have a positive impact on well-being [10,11,12].

#### 1.1.2. Gender Differences in Employment

Stable employment is influenced by both individual factors and workplace environmental factors. Prior research has identified ten factors that are common across all types of organizations—job satisfaction, job stress, organizational culture, organizational commitment, salary, organizational justice, opportunities for promotion, demographic variables, leadership style, and organizational climate—as well as additional factors that warrant consideration [17]. In recent years, there has been a growing focus on turnover factors among younger generations. Regarding individual factors, emotional exhaustion, job satisfaction, and passion for work have been suggested as important [18,19]. With respect to workplace environmental factors, recent findings suggest that the balance with compensation may play a critical role [20]. Thus, in the pursuit of stable employment, it is presumed that many individuals consider these factors to be important already at the point of job-hunting. Job-hunting may carry diverse meanings across countries and cultures due to differences in educational systems; however, it is an important activity for achieving stable employment continuity [15]. In Japan, employment is generally categorized into two main tracks: recruitment of new graduates and mid-career hiring. The process often involves career counseling at educational or public institutions, participation in company information sessions, preparation and submission of application documents such as résumés, and attendance at job interviews, eventually leading to an informal job offer or formal employment [15]. In this study, we operationally define this series of steps leading up to employment as ‘job-hunting’.

Furthermore, prior research examining determinants of employee retention across different generations suggests that work–life balance may be important for all generations [21]. Studies focusing on younger workers indicate that high compensation alone is insufficient, and that life satisfaction may also be an important factor [20]. Work–life balance has also been found to be associated with gender role orientation [22,23]. In particular, conflicts arising in situations where maintaining work–life balance is difficult have been reported to be stronger among women with egalitarian values than among those with traditional values [23]. Therefore, it has been pointed out that gender roles associated with gender itself may be a more important factor affecting work-life balance than gender itself [22]. Women are entering the labor force, and gender equality is increasing worldwide. The United Nations (UN) adopted the Convention on the Elimination of All Forms of Discrimination against Women in 1979 at the 34th session of the UN General Assembly [24]. Japan followed suit, ratifying the Convention on the Elimination of All Forms of Discrimination against Women in 1985 and enacting the Equal Employment Opportunity Law the following year [25]. The enactment of this law greatly reduced gender inequality in terms of employment in Japan and narrowed the employment rate gap between men and women [26]. However, even when women find employment, they face poor working conditions [27]. Prior research suggests that paid work may negatively affect women’s well-being in regions with a strong sense of gender role division of labor, such as East Asia [28]. However, the limitation of these studies is their insufficient consideration of gender perspectives [29].

#### 1.1.3. Social Role Theory

Gender role orientation is conceptualized within Social Role Theory, which explains the origins of gender differences in a wide range of social behaviors [30,31]. Traditional gender roles refer to the division of labor in which men are expected to engage in paid work, while women are expected to undertake household labor [30,31]. According to this theory, psychological differences between men and women stem from the division of labour and roles within society. In other words, it emphasizes that such differences are not solely the outcome of universal evolutionary processes but are strongly shaped by social structures. Recent research further indicates that attitudes toward traditional gender roles are associated with individual personality traits and preferences regarding full-time homemaking [32]. Japan continues to lag significantly behind other countries in terms of gender equality, and such societal conditions may influence young people entering the workforce [33]. In other words, it is not yet clear whether traditional gender role attitudes constitute an important factor for well-being during the job-hunting stage. Drawing on prior studies that have examined factors related to both retention and turnover, the present study proposes the following hypothesis.

**H1.** 
*Among Japanese emerging adult women, job hunting is positively related to well-being.*


**H2.** 
*For emerging adult women who hold traditional gender role attitudes, job hunting is associated with lower well-being.*


The high number of Japanese emerging adult women leaving the workforce may be influenced by their awareness of gender roles in the division of labor. Arnett said that “the rise in the ages of entering marriage and parenthood, the spread of the education and training beyond secondary school, and prolonged job instability during the 20s reflect the rise of a new life stage for young people” [1]. Notably, few studies have focused on emerging adults, whose demographics have evolved with changing times and diverse lifestyles. Job-hunting is an important first step in the transition to the working stage. Therefore, this study aimed to clarify the relationship between the attitude toward gender roles, job-hunting experiences, and the well-being of emerging adult women. Stable and continuous employment among younger generations of workers is important not only for the individuals themselves but also for society as a whole. In particular, women’s active participation in the economic sphere is increasingly expected; however, factors influencing stable employment among young women remain insufficiently understood. Gender role orientation may be one factor that helps fill this gap. In a society that is promoting gender equality, yet in which traditional expectations and gender role norms persist in everyday life, focusing on the values of young women is essential for supporting the contributions of the next generation.

## 2. Methods

### 2.1. Participants and Procedure

Participants were recruited using purposive sampling. This study was conducted as part of the Co-creation Opportunity Formation Support Program (COI-NEXT; FY2022–2031). It focused on younger generations, including emerging adults (aged 18 to 29), the age group with the highest turnover rate in Japan. The participants in this study were women selected from individuals who participated in the program. Recruitment for this study was conducted at five universities and five companies, mainly located in Yokohama and Tokyo. The universities and companies targeted in this study were organizations participating in the COI-NEXT. Yokohama and Tokyo are regions that attract a large influx of young people from across Japan. A cross-sectional online survey was conducted among emerging adult women between July and September 2023. The web-based questionnaire was administered using Google Forms. Each university and company utilized various communication tools (such as portal sites, web bulletin boards, and notice boards) to post recruitment materials and leaflets. The inclusion criteria for this study were the ability to read and write Japanese, while the exclusion criterion was the absence of informed consent.

### 2.2. Measures

Well-being: We measured overall well-being using the Japanese version of the PERMA-Profiler [34,35]. According to previous studies, well-being consists of three main elements: hedonic, eudemonic, and evaluative well-being [36]. The PERMA model used in this study incorporates components of hedonic and eudemonic well-being [16]. It has been translated into various languages and is used worldwide [37]. In this study, we used overall PERMA well-being as an outcome. The Cronbach’s alpha for this study was 0.937.

Job-hunting: This was defined as any life event experienced within the past 6 months that resulted in getting a job or job search, such as participating in an internship or job interview, using a two-choice, single-answer question.

Traditional gender-role attitudes: This was measured using a gender role stressor scale [38]. This consisted of 12 questions, such as “Women are not expected to express their opinions clearly in work and other contexts,” and “Even in the same position, women are not trusted, with 100% of the responsibilities assigned to men.” These questions were answered on a scale of 1 to 5, ranging from 1 = “I don’t feel very uncomfortable at all” to 5 = “I feel very uncomfortable.” This scale range was 12–60, with lower scores indicating stronger traditional values. The reliability and validity of this scale were confirmed on female workers in Japan [38]. The Cronbach’s alpha for this study was 0.947. In this study, the total score was divided into two median values, which were used as interaction terms.

Covariates included age, occupation (student/working adult), household situation (living alone/living with family/other), present illnesses (none/one or more), self-assessed living conditions (very comfortable/somewhat comfortable/normal/somewhat difficult/very difficult), personality (extraversion/agreeableness/conscientiousness/neuroticism/openness to experience), and perceived social support. These factors were included as covariates, as previous studies have suggested that they may confound the associations between job hunting and well-being, as well as between traditional gender role attitudes and well-being [39,40,41,42,43,44,45,46,47].

### 2.3. Statistical Analysis

The dependent variable was well-being, and the explanatory variable was job-hunting experiences. The interactional variable was traditional gender-role attitudes. A multiple regression analysis was conducted using the interaction terms of the explanatory variables while controlling for covariates. Statistical analyses were conducted using Stata software, BE 18.0 (StataCorpLLC, College Station, TX, USA). The level of statistical significance was set at *p* < 0.05.

### 2.4. Patient and Public Involvement Statement

Neither the patients nor the public were involved in designing the study or disseminating the study results.

### 2.5. Ethics Approval and Consent to Participate

This study was approved by the Ethics Review Committee of Yokohama City University (F230200064). Prior to participation, respondents were informed that the survey was anonymous, self-reported, and voluntary, and that they could withdraw at any time without penalty. At the outset of the online questionnaire administered via Google Forms, they were also informed that all data would be treated confidentially and used solely for academic and scientific purposes. Submission of the completed questionnaire was taken as an indication of informed consent for participation and data processing.

## 3. Results

Of the 137 potential participants, data from 132 women in emerging adulthood with complete responses were analyzed, including 35 who had job-hunting experience. Of these, 11 were employed. Table 1 shows participant characteristics. The mean score for outcome well-being was 6.10 (Standard Deviation ([SD]) = 1.61). The mean score for traditional gender-role attitudes was 53.11 (SD = 8.69). Multiple regression analysis demonstrated a negative relationship between job-hunting experience and well-being (Table 2). The Partial regression coefficient B was −0.81 (95% CI: −1.52, −0.10).

Figure 1 illustrates that individuals with more traditional gender-role attitudes had lower levels of well-being when under employment or having job-hunting experiences. The variance inflation factor was <5. The results of the correlation analysis between well-being and each variable are presented in Appendix A.

## 4. Discussion

The findings did not provide support for Hypothesis 1, while partial support was observed for Hypothesis 2. While the target demographics of previous studies involving Japanese workers differed slightly from that of this study, compared to the well-being score of a previous study (mean score of overall well-being: 5.88), the well-being of the current group was somewhat higher [35]. The participants in previous studies were, on average, in their 40s, which is older than the participants in this study, and they represented a diverse group in terms of job types and educational backgrounds. This study revealed a negative association between job-hunting experience and well-being, particularly in emerging adult women with more traditional attitudes toward gender roles. This indicates that the relationship between job-hunting experience and well-being becomes increasingly negative as traditional attitudes toward gender roles strengthen.

In terms of how job-hunting and gender role attitudes interact, prior studies clarified that there’s no noticeable difference in how Generation X (born between 1965–1980) and Generation Z (born between 1997–1912) perceive gender roles [48]. Despite advocacy for gender equality in employment-related legislation, the active participation of women in the economic sector, as evidenced by Japan’s gender gap index, remains limited [33]. Consequently, women planning to assume significant household responsibilities may struggle to balance their work and domestic duties. Currently, life events such as marriage and childbirth are the main reasons women in their 20s and 30s leave their initial jobs [49]. A prospective occupational cohort study in the United Kingdom found that women were more likely to leave their jobs because of conflicts between work and family [50]. This study also considered individual personality traits and yielded findings consistent with prior research on preferences for full-time homemaking [32]. In the Japanese economic context, the persistent gender gap may help to explain job-hunting behaviors among emerging adult women who have been educated in a more gender-equal society, and the present results align with the propositions of Social Role Theory [30,31]. Thus, employment may not necessarily improve the well-being of new adult women who seek to remain loyal to their traditional roles. Conflicts may arise when the role expected of those around them differs from the values the individual has for that role [51]. A longitudinal study of women found that declining social roles were associated with an increase in mental illness [52]. Previous studies in East Asia have indicated that tailored interventions and smooth transition support are important for improving the young individual’s job-hunting experience [53,54]. The findings of this study underscore the significance of accounting for emerging adult women’s perspectives on traditional gender roles, along with other factors considered.

The results suggest that job-hunting tends to decrease well-being by −0.81. Most participants in this study were involved in a job search process. Consequently, most of them work at their current jobs with a lot of dissatisfaction, or since their employment status was not yet determined, they did not experience the by-products of employment, such as financial rewards and social connections, which might have otherwise contributed to an increase in well-being. Previous research has suggested that unstable employment may have detrimental effects on mental health [9,10,11]. Concerning the impact of job-hunting on well-being, both employment and job-hunting can potentially boost the well-being of emerging adult women, in line with previous research [3]. This study focused on women. Previous research has suggested that, for women, social support and psychological resilience may be more strongly associated with life satisfaction [55]. Accordingly, supportive interventions targeting emerging adult women could potentially enhance their life satisfaction, which in turn may foster more positive feelings toward job hunting. Through job-hunting, these women can develop a vision for their careers and enhance their sense of well-being through successful experiences.

## 5. Strengths and Limitations

This study explored the link between gender-role attitudes, job-hunting, and well-being among Japanese emerging adult women, revealing interesting trends. However, this study had some limitations.

First, ambiguities may exist in the definition of employment. This study did not collect data on the number of job interviews or the duration of job-hunting activities. Participants in this study were asked whether they had been employed or engaged in job hunting within the past six months. In cases where participants did not respond to the job-hunting question but indicated that they had been employed, we classified them as having likely engaged in job hunting followed by employment. In addition, an individual who persistently seeks employment without success may experience diminished well-being. The employment status of the participants, such as part-time jobs, was also unclear. Second, differences in gender and professions were not clarified. Therefore, gender differences were not discussed. Moreover, for working adults in their late 20s, the job search might have been for a career change; thus, the mechanisms could have been different. Third, this study targeted emerging adults living in relatively urban areas in Japan. Furthermore, populations with higher levels of well-being may have responded to this study. Therefore, the respondents in this study may have represented a cohort with a marginally higher level of well-being, thereby limiting its generalizability. Finally, because this study was cross-sectional, it could not address the causal relationship between job-hunting and well-being. Finally, because this study employed a cross-sectional design, causal inferences cannot be drawn. To more clearly examine the association between job hunting and well-being among emerging adults, as well as the moderating role of traditional gender role orientation, future research should adopt longitudinal designs and consider gender role orientations across different generations, genders, and cultural contexts.

## 6. Conclusions

This study provides preliminary indications that job-hunting experiences may be linked to well-being in different ways depending on women’s gender role attitudes. For example, the results suggest a possible tendency for women with more egalitarian views to report higher levels of well-being, whereas those with more traditional attitudes may report lower levels. However, because the sample consisted only of emerging adult women living in urban areas, these findings should be interpreted with caution and cannot be generalized to all populations.

Future studies should utilize larger sample sizes to enhance the comprehension of the distinctions between employment and job-hunting experiences, as well as the potential variances in mechanisms across diverse attributes.

## Figures and Tables

**Figure 1 ijerph-22-01385-f001:**
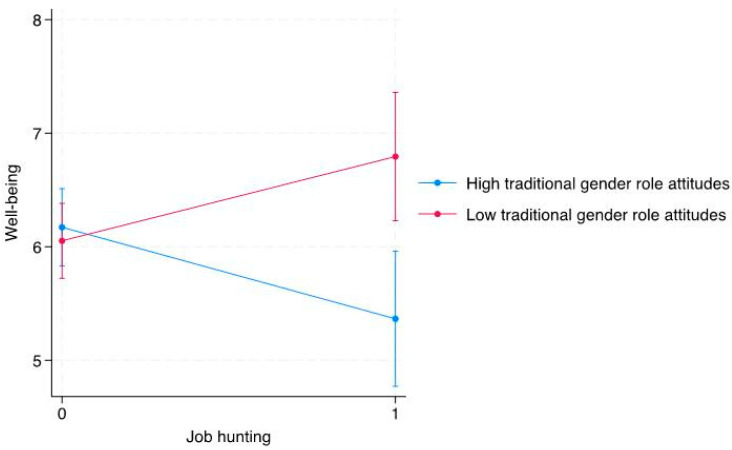
Interaction between employment, job hunting, and gender role attitudes in predicting well-being scores, with 95% Confidence Interval.

**Table 1 ijerph-22-01385-t001:** Participant characteristics (N = 132).

		n [%] or Mean [SD]	
Age (range:18–29)		24.23	[3.23]
Occupation	Student	40	[30.30]
	Working adult	92	[69.70]
Household situation	Living alone	47	[35.61]
	Living with family	75	[56.82]
	Other	10	[7.58]
Present illness	None	66	[50.00]
	One or more	66	[50.00]
Self-assessed living conditions	Very comfortable	10	[7.58]
	Somewhat comfortable	29	[21.97]
	Normal	65	[49.24]
	Somewhat difficult	22	[16.67]
	Very difficult	6	[4.55]
Personality	Extraversion	8.11	[3.01]
	Agreeableness	10.29	[2.41]
	Conscientiousness	6.73	[2.70]
	Neuroticism	9.08	[2.66]
	Openness to experience	8.07	[2.84]
Perceived social support (range: 7–49)		39.01	[8.17]
Well-being (range: 0–10)		6.10	[1.61]
Job-hunting	No	97	[73.48]
	Yes	35	[26.52]
Gender role attitudes (range:12–60) *		53.11	[8.69]

SD, standard deviation * Lower scores indicate stronger traditional values.

**Table 2 ijerph-22-01385-t002:** Association between job-hunting and well-being (multiple regression analysis).

	Crude			Adjusted		
	B	95% CI	*p*-Value	B	95% CI	*p*-Value
1 Job-hunting	0.18	[−0.46–0.81]	0.58	−0.81	[−1.52–−0.10]	0.03
2 Gender-role attitudes				−0.12	[−0.60–0.36]	0.62
1 × 2 interactions				1.55	[0.58–2.51]	0.00
Age				0.06	[−0.02–1.43]	0.15
Occupation				0.82	[0.21–1.43]	0.01
Household situation						
Living alone (ref.)				-	-	-
Living with family				0.14	[−0.29–0.57]	0.52
Other				−0.19	[−1.00–0.62]	0.64
Present illness				0.30	[−0.11–0.71]	0.16
Self-assessed living conditions						
Very comfortable (ref.)				-	-	-
Somewhat comfortable				−0.70	[−1.56–0.16]	0.11
Normal				−0.42	[−1.23–0.39]	0.31
Somewhat difficult				−0.85	[−1.76–0.05]	0.06
Very difficult				0.21	[−1.05–1.46]	0.74
Personality						
Extraversion				0.04	[−0.03–0.12]	0.26
Agreeableness				0.02	[−0.07–0.11]	0.60
Conscientiousness				0.07	[−0.01–0.15]	0.08
Neuroticism				−0.13	[−0.22–−0.04]	0.00
Openness to experience				0.09	[0.02–0.17]	0.00
Perceived social support				0.11	[0.09–0.14]	<0.00
R^2^	0.00			0.52		

B: partial regression coefficient, 95% CI: 95% confidence interval, R^2^: coefficient of determination adjusted for degrees of freedom.

## Data Availability

The data used in the present study are kept securely by Yokohama City University, and no data will be publicly shared. Any queries relating to data access can be made to: ycu.minds.t1@yokohama-cu.ac.jp.

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
