# Peer review of "Traditional Gender Role Attitudes and Job-Hunting in Relation to Well-Being: A Cross-Sectional Study of Japanese Women in Emerging Adulthood"

_ijerph, 2025, doi:10.3390/ijerph22091385_

Round 1
Reviewer 1 Report
Comments and Suggestions for Authors
First, I want to thank the authors for their impressive work. Based on the review, I’ve shared several observations for further improvement below.
- Please revise the abstract according to journal guidelines. A standard abstract should not include demographic information, conclusion, or the strengths and limitations of this study.
- The study fails to adequately address the research gap, a crucial aspect of any empirical study. Moreover, a lack of studies does not necessarily indicate a research gap. What makes your research or framework novel compared to existing work? And why is your study important for your audience?
- In lines 88-89, it was stated that “Notably, few studies have focused on emerging adults, whose demographics have evolved with changing times and diverse lifestyles”. Please add citations to justify a few studies. Again, “Job-hunting is an important first step in the transition to the working stage”. How can you make that claim without reviewing the literature? Please justify.
- This investigation lacks a literature review section. How can the audience understand the influence of study variables such as attitude toward gender roles, job-hunting experiences, and well-being?
- Please provide more details on the data collection process (survey instrument distribution, start and end of data collection, pilot test, etc.).
- For better clarity, please describe the study population so readers can justify the sample size. Could you please explain the rationale for selecting only five universities and five companies, primarily located in Yokohama and Tokyo?
- What is the reason for choosing respondents aged 18–29 years?
- Please specify the sampling method used to select respondents.
- No theoretical and practical implementations were provided, which are crucial for any empirical study and also vital for the organizations.
Best wishes.
Comments on the Quality of English LanguageThe English could be improved to more clearly express the research.
Reviewer 2 Report
Comments and Suggestions for Authors
- In the abstract, the relationship between attitudes towards traditional gender roles, job search, and the well-being of young adult women in Japan remains unclear, indicating a lack of understanding of these dynamics. This serves as a sufficient introduction to the problem.
- In the abstract section, an online survey-based cross-sectional study was conducted at universities and companies in Yokohama and Tokyo, with a female population aged 18-29. It would be beneficial to include the number of samples and the primary measuring tools used (e.g., a specific psychometric scale).
- Some parts seem less directly connected, so they need to be rearranged to clarify the relationship between ideas.
- The introduction effectively highlights the importance of topics, such as the relationship between work and the well-being of young adults in Japan, as well as the existence of gaps in research on gender attitudes and job search. However, the placement of international contexts and related literature is still less prominent. It would also be beneficial to introduce the theoretical framework on which the research is based.
- It is recommended that the introduction begin with an overview of the importance of research in the global realm, particularly in Japan, and then focus on specific issues related to gender, job search, and well-being.
- The unclear relationship between traditional gender attitudes, job search, and the well-being of young adult women in Japan, as well as the lack of research examining these interactions specifically in the context of Japanese culture.
- The introduction follows a general funnel pattern, beginning with an overview of young adult employment and well-being, and then addressing specific issues related to gender and job search. However, the transitions between paragraphs can be smoothed out to make them more flowing.
- To strengthen the author’s argument, he needs to affirm what literary void he wants to fill explicitly.
- The data collection procedure, participant selection process, and statistical analysis steps are described in more detail, including measurement parameters, data collection duration, and inclusion/exclusion criteria.
- Details on cultural adaptation and validation of instrument reliability in Japanese samples are included, including relevant Cronbach’s alpha values as part of the methodology reference.
- It is essential to emphasize that the assumption of cause-and-effect relationships should be carefully conveyed, given the cross-sectional study design.
- Longitudinal experiments are also highly recommended to overcome the limitations of cross-sectional studies and ensure cause-and-effect relationships.
- This conclusion should be avoided, as it would lead to overly general claims without considering methodological limitations, such as the cross-sectional nature of the study and the limited sample size in urban areas of Japan.
Reviewer 3 Report
Comments and Suggestions for Authors
Dear Authors,
Your study brings up an interesting research question about women’s unique career experiences and well-being in Japan. I also appreciate the effort you’ve put in collecting the data across multiple institutions and industries. In the meantime, I also want to share several observations of the manuscript with you, and I hope they are helpful.
- Definitions of Constructs:
There are many concepts (constructs/variables) in your writing, yet none of them is explicitly defined. It is difficult for readers to grasp what you mean by them and therefore not clear why you think they are supposed to be related in a particular way.
For instance, “job hunting” – the writing seems to mix these thing together, employment, find employment, job hunting, job hunting experience, job searching, getting a job, being employed. But they are not the same especially job hunting vs getting a job vs having a job. The distinctions among these terms mean they are very different experiences and will lead to very different outcomes in terms of well-being.
- Specify Hypotheses:
Related to the muddy conceptualization of the key constructs/variables, the writing never explicitly specify what kind of relationships you are predicting. A good job hunting experience can lead to higher well-being while a bad job hunting experience will lead to lower well-being. Having a job comparing with not having a job might lead to higher well-being. And more importantly, having a job that you are happy with will positively affect well-being. But none of these nuances are clearly articulated.
More importantly, it is important to provide the “why” underlying the hypothesis statements, a theory that provides explanations for the relationships between variables.
Without these steps, it will be difficult to judge if your statistical analysis is appropriate and if your interpretation of the data is sound.
Comments on the Quality of English LanguageConceptual clarity needs to be improved in the writing.
Round 2
Reviewer 1 Report
Comments and Suggestions for Authors
All the comments have been addressed appropriately. I have no other concerns. Thank you.
Author Response
We are grateful for the opportunity to refine our study and would like to express our sincere appreciation for your thoughtful and thorough review despite your busy schedule.
Reviewer 3 Report
Comments and Suggestions for Authors
Dear authors,
Thank you for your efforts to revise the manuscript. I appreciate the explicit hypothesis statements. I am afraid the issues I raised have not been fully addressed yet.
- Definitions of variables: there are no definitions. You are listing all the concepts (as factors) in the literature without defining them, without articulating why they are relevant to your research question.
- Theoretical foundation: the role of a theory is to help you explain why you think certain variables are related to each other in certain way. In fact, role theory, as you articulated, actually would predict different relationships as your hypotheses have stated.
- The data serve the goal of testing your hypotheses. But you seem to collect the data very randomly because you did not specify in your hypothesis development section why and how the variables in your data might be relevant to address your research question. The analysis is disconnected from your hypotheses.
In summary, your manuscript lacks of a logical coherence among these key components of an empirical paper: research question, theoretical foundation (including variable selection and definitions), hypothesis statements, data analysis. In many ways, each component is very under development. As a result, they are disconnected from each other.
Comments on the Quality of English LanguageModerate English editing is needed to help improve the clarity of the writing.
